# Emergent Molecular Techniques Applied to the Detection of Porcine Viruses

**DOI:** 10.3390/vetsci10100609

**Published:** 2023-10-07

**Authors:** Elda A. Flores-Contreras, Jorge Alberto Carrasco-González, Daniel C. L. Linhares, Cesar A. Corzo, J. Israel Campos-Villalobos, Alexandra Henao-Díaz, Elda M. Melchor-Martínez, Hafiz M. N. Iqbal, Reyna Berenice González-González, Roberto Parra-Saldívar, Everardo González-González

**Affiliations:** 1Tecnologico de Monterrey, School of Engineering and Sciences, Monterrey 64849, Nuevo Leon, Mexico; eldafc@tec.mx (E.A.F.-C.); elda.melchor@tec.mx (E.M.M.-M.); hafiz.iqbal@tec.mx (H.M.N.I.); 2Institute of Advanced Materials for Sustainable Manufacturing, Tecnologico de Monterrey, Monterrey 64849, Nuevo Leon, Mexico; 3Biosafety, Lomas de los Pinos No. 5505—F, Col. La Estanzuela vieja, Monterrey 64984, Nuevo Leon, Mexico; jorge.carrasco@biosafety.mx; 4Veterinary Diagnostic and Production Animal Medicine Department, College of Veterinary Medicine, Iowa State University, Ames, IA 50011, USA; linhares@iastate.edu; 5Veterinary Population Medicine Department, College of Veterinary Medicine, University of Minnesota, St. Paul, MN 55455, USA; corzo@umn.edu; 6Proteina Animal S.A. de C.V. (PROAN), San Juan de los Lagos 47000, Jalisco, Mexico; israel.campos@proan.com; 7Industrias Bachoco, S.A. de C.V., Celaya, Celaya 38010, Guanajuato, Mexico; alexandra.henao@genusplc.com

**Keywords:** PCR, LAMP, RPA, NASBA, PSR, ASFV, PRRSV, PCV, PEDV

## Abstract

**Simple Summary:**

In this review, we examine the application of different molecular diagnostic techniques in the pig industry; qPCR, isothermal methods, and novel techniques such as CRISPR-Cas and microfluidics platforms are discussed in detail. The main viruses affecting the health of pigs were identified, including the African swine fever virus, porcine reproductive and respiratory syndrome virus, porcine epidemic diarrhea virus, and porcine circovirus. Furthermore, the main challenges for the implementation of large-scale molecular diagnostic tests in the swine industry are discussed, which include lower costs, ease of operation for any type of user, portability for use on farms, and rapid response times.

**Abstract:**

Molecular diagnostic tests have evolved very rapidly in the field of human health, especially with the arrival of the recent pandemic caused by the SARS-CoV-2 virus. However, the animal sector is constantly neglected, even though accurate detection by molecular tools could represent economic advantages by preventing the spread of viruses. In this regard, the swine industry is of great interest. The main viruses that affect the swine industry are described in this review, including African swine fever virus (ASFV), porcine reproductive and respiratory syndrome virus (PRRSV), porcine epidemic diarrhea virus (PEDV), and porcine circovirus (PCV), which have been effectively detected by different molecular tools in recent times. Here, we describe the rationale of molecular techniques such as multiplex PCR, isothermal methods (LAMP, NASBA, RPA, and PSR) and novel methods such as CRISPR-Cas and microfluidics platforms. Successful molecular diagnostic developments are presented by highlighting their most important findings. Finally, we describe the barriers that hinder the large-scale development of affordable, accessible, rapid, and easy-to-use molecular diagnostic tests. The evolution of diagnostic techniques is critical to prevent the spread of viruses and the development of viral reservoirs in the swine industry that impact the possible development of future pandemics and the world economy.

## 1. Introduction

In recent years, we have experienced significant technological advances to combat diseases. The primary focus has been on human health; however, recent pandemics have evidenced the urgent need for monitoring and controlling diseases in the animal sector, such as on swine farms, because swine farms play a vital role in human health and nutrition, as well as promoting rural poverty reduction. In addition, pigs are prone to developing reservoirs of potential pathogens that could lead to pandemics in humans via cross-species transmission [1].

In the swine sector, there has been a noteworthy development based on vaccines and diagnostic tests to control diseases. The global population of pigs was estimated to be above 784 million in 2022; China alone contains more than half of the total global pig population [2]. Nevertheless, as disease prevention efforts have not been sufficient, there is still an ongoing global threat from swine viruses. Moreover, the improper management of diseases in the swine industry affects not only human and animal health issues but also socioeconomical aspects. For example, the disease caused by the porcine reproductive and respiratory syndrome virus (PRRSV) leads to losses of more than USD 500 million annually and is considered the most expensive disease faced by pig producers [3]. In this context, pork production is severely affected by inadequate disease management. Figure 1 shows the main swine viruses that impact porcine farms and the world’s swine population; although these viruses have low or no zoonotic potential, it is important to use molecular diagnostic tools to prevent zoonoses and the development of pandemics.

To achieve the objective of health safety, substantial effort is needed, mainly toward massifying molecular tests. Reverse transcription polymerase chain reaction (RT-PCR) and real-time PCR (qPCR) are among the main molecular diagnostic tests that are used on a large scale to detect swine viruses and are considered the gold standard due to their high sensitivity and specificity [4]. Despite the effectiveness of these techniques for virus detection, their use is costly and limited to specialized laboratories with highly qualified staff [4]. In this context, molecular diagnostic tests must be easily accessible in any part of the world that can potentially affect the world population through the presence of pathogens. Many countries cannot afford routine diagnostic expenses; thus, it is crucial to drastically reduce the costs of diagnostic techniques. To do this, the tests should not require sophisticated equipment or highly qualified personnel. More importantly, early field diagnosis is considered the first line of defense; thus, enhanced diagnostic measures lead to effective disease prevention [5]. In this sense, new molecular diagnostic tests have been developed, such as microfluidics assays that allow the setting up of a laboratory on a chip. The new assays provide a large-scale and precise molecular diagnosis. Isothermal tests such as loop-mediated isothermal amplification (LAMP), nucleic acid sequence-based amplification (NASBA), recombinase polymerase amplification (RPA), and polymerase spiral reaction (PSR) have also been proposed, which allow quick results with high sensitivity and specificity and without using thermal cyclers to detect the presence of viruses [6].

In this review, we aim to compile the recent developments in diagnostic tests for detecting porcine viruses. We discuss the most common methods such as PCR, which has been extensively used over the years, and other emerging molecular techniques that can be used for the early detection of porcine viruses that cause significant economic losses worldwide. In addition, these techniques can allow quick implementation of security measures to prevent the spread of potentially harmful viruses developing into pandemics.

## 2. Main Swine Viral Diseases

African Swine Fever Virus (ASFV)

The ASFV is the agent responsible for African swine fever (ASF); it was identified in Kenya in 1921 and has a mortality rate of up to 100%. The virus belongs to the *Asfarviridae* family, and its genome is made up of double-stranded DNA with a size of 170 to 193 kb and encodes more than 50 proteins [7]. ASFV is a very stable virus in the environment; the half-life of infection in feces can be 0.41 days at 37 °C [8]. The clinical manifestation varies, mainly depending on the route of infection and the viral dose of exposure; common symptoms include loss of appetite, temperatures above 40 °C, inactivity, pulmonary edema, and hemorrhages [9].

Molecular diagnosis of ASF is vital due to its similarity with other diseases, such as classical swine fever (CSF), erysipelas, and septicemic salmonellosis. Samples recommended for diagnosis include blood, serum, lymph node, kidney, spleen, and lung. Disease diagnosis can be achieved by detecting antibodies or viral antigens, in which serological tests, such as variants of ELISA, can be performed [10]. However, the detection of nucleic acids is based on qPCR, which is the diagnostic method currently recommended by the World Organization for Animal Health (OIE) [10]. It is essential to note that there are no available effective treatments or vaccines for ASF; therefore, the best approach to combat the disease is through epidemiological surveillance and molecular diagnosis strategies.

Porcine Reproductive and Respiratory Syndrome Virus (PRRSV)

The PRRSV is responsible for porcine reproductive and respiratory syndrome (PRRS). This name was assigned in 1991 by Terpstra et al. after spending years observing a pattern of clinical signs in pigs [11]. This virus belongs to the *Arteriviridae* family and is divided into PRRSV1 (European) and PRRSV2 (North American); the species differ by 44% of their genomes [9]. The PRRSV genome comprises RNA with an approximate size of 15.4 kb, with 10 open reading frames (ORFs) that code for seven structural proteins [9].

PRRS causes abortions, premature or late deliveries, and stillborn or weak piglets, which have negative effects on pig production [9]. Regarding clinical manifestations, some common symptoms include fever, dyspnea, apathy, and anorexia. Moreover, younger pigs are more severely affected by this virus than adults [9]. The transmission of PRRSV can be via intramuscular, intranasal, oral, vaginal, or intrauterine routes. It has been reported that PRRSV exhibits good stability at pH values in the range between 6.5 and 7.5 [12].

There are also serological and nucleic acid tests for PRRSV diagnosis that can be used in conjunction with other methods to obtain as much clinical field information as possible. Currently, there are a variety of vaccines for PRRS; therefore, the adequate monitoring of the production of anti-PRRSV antibodies is important and is carried out using ELISA tests. On the other hand, since it is an RNA virus, it is necessary to use RT-qPCR, which is recommended by the OIE since it is one of the most sensitive techniques to detect genetic material [10].

Porcine Epidemic Diarrhea Virus (PEDV)

Although Debouck et al. named the PEDV virus that causes porcine epidemic diarrhea (PED) in 1982, there have been reports of significant PED-like effects since 1970 [9]. This virus belongs to the *Coronaviridae* family; its genome is composed of RNA with a size of approximately 28 kb that contains seven ORFs that code for 4 structural proteins and 17 non-structural proteins [9].

PEDV is one of the primary enteric pathogens that affect pigs, and the clinical manifestation of PED is accompanied by cases of watery diarrhea, vomiting, anorexia and depression [13]. ELISA and RT-qPCR tests have been developed to diagnose this disease that are similar to those used to diagnose ASFV and PRRSV. Nevertheless, novel molecular diagnostic methodologies, such as LAMP, have recently been proposed [13].

Porcine Circovirus (PCV)

PCV, which belongs to the *Circoviridae* family, is a DNA virus with an approximate size of 1.7 to 2.0 kb containing two main ORFs responsible for coding for the Rep proteins (ORF1) and the viral capsid structural protein Cap (ORF2). Currently, four species have been identified: PCV1, PCV2, PCV3, and PCV4. PCV1 was first identified in 1974 and is considered non-pathogenic [14]; PCV2 was recognized as the cause of porcine circovirus diseases (PCVDs) in 2012 and continues to be widely studied [11]; PCV3 and PCV4, which were discovered most recently, were found in 2016 and 2020, respectively [15,16].

Regarding diagnosis, PCV2 has generated and greatly influenced important technological development advancements in both areas of diagnosis and treatment. For example, there are currently five commercial vaccines to prevent PCV2 and modern molecular diagnostic tools that allow this virus to be identified directly on farms using isothermal and microfluidics techniques.

## 3. Multiplex qPCR

The PCR technique, created almost four decades ago (in the year 1983), is a molecular biology tool that has revolutionized the history of humanity and has been widely used in disease diagnosis. It should be noted that different PCR variants exist. Among them, quantitative PCR (qPCR) and quantitative reverse transcription PCR (RT-qPCR) are the most commonly used methods. In addition, during the COVID-19 pandemic, RT-qPCR was used as never before (despite the high cost for some population sectors) owing to its high sensitivity and rapid detection compared with other techniques. Overall, qPCR provides significant advantages over the classic PCR endpoint, such as increased sensitivity and the ability to quantify fluorescence signals when using specifically designed probes labeled with fluorophores [17]. For this reason, the authorities have recommended using qPCR as the gold standard for many diseases that affect humans, animals, and plants [10,18,19,20].

Since its development, PCR has been considered a tool for detecting more than one target or amplicon in the same reaction, generating a multiplex PCR capable of detecting several fragments of genetic material [21]. Currently, several multiplex PCR assays are being developed by research groups and companies to identify viruses that affect pigs. Thus, multiplex PCR has great potential for disease diagnosis because, in a reaction where a single target is traditionally identified, up to five targets can now be identified [22]. In this manner, the objective of a multiplex PCR is to make the most of resources, mainly in terms of time and reagents.

In this section, we discuss RT-qPCR and qPCR multiplex tests. However, it should be stated that there are also multiplex endpoint PCR tests that require agarose gel electrophoresis to identify the presence of amplicon bands. Some of the viruses that have been detected through this approach are: the African swine fever virus (ASFV), classical swine fever virus (CSFV), porcine reproductive and respiratory syndrome virus (PRRSV), porcine pseudorabies virus (PRV), porcine circovirus types 1–3 (PCV1–PCV3), porcine rotavirus A (PoRV-A), porcine epidemic diarrhea virus (PEDV), transmissible gastroenteritis virus (TGEV) and getah virus, porcine parvovirus (PPV), Japanese encephalitis virus (JEV), porcine astroviruses 1–5 (PastV1–PastV5), and PPV (PPV1~PPV7) [23,24,25,26,27,28,29,30,31,32].

The first attempt to develop a qPCR multiplex was proposed by Zheng et al., where they reported a triplex qPCR test for detecting the PPV2, PRV, and PCV2 viruses, using SYBR Green I to identify pathogens. The authors declared limits of detection (LODs) of 14 copies for PPV, 14 copies for PRV, and 10 copies for PCV2 [33]. One of the most interesting proposals for multiplex qPCR tests was that from Chen et al., as theirs made it possible to simultaneously identify three of the most important viruses that affect pigs. Triplex RT-qPCR detected the ASFV (p72 gene), CSFV (5′ UTR), and PRRSV (ORF7) viruses, which were recognized by specific probes labeled with FAM, VIC, and Cy5 fluorophores, respectively. The authors reported an LOD of 1.78 copies for all viruses [34]. A similar triplex RT-qPCR was developed by a research group in China, detecting two of the same viruses, ASFV (p72) and CSFV (5′ UTR); however, the test could also identify atypical porcine pestivirus (APPV) by amplifying the 5′ UTR genomic region. The test worked through probes labeled with the Texas Red, JOE, and FAM fluorophores, respectively. The authors reported an LOD of 2.52 copies for all viruses [35].

Recently, four research groups have developed quadruplex tests to detect different viruses. The first was the quadruplex RT-qPCR proposed by Zhu et al., which was designed for the detection of PEDV (N gene), porcine deltacoronavirus (PDCoV) (N gene), TGEV (N gene), and swine acute diarrhea syndrome coronavirus (SADS-CoV) (S gene), using probes labeled with JUN, ABY, VIC, and FAM, respectively. The authors validated the test with 1807 clinical samples and determined LODs of 8 copies for PEDV, 4 copies for PDCoV, 16 copies for TGEV, and 6.8 copies for SADS-CoV [22].

The second test was able to detect the same viruses but its design was based on other targets and used different dye reporters: PEDV (N gene-JOE), PDCoV (M gene-FAM), TGEV (N gene-Texas Red), and SADS-CoV (N gene-Cy5). This test was validated using 3236 clinical fecal samples and demonstrated an LOD of 121 copies/μL for each virus [36].

The third quadruplex was presented by Pan et al. and was capable of detecting PEDV, PDCoV, porcine torovirus (PtoV), and SADS-CoV. The authors designed specific probes for viral genes and dye reporters including the PEDV-FAM-ORF1a gene, PDCoV-Texas Red-ORF1b gene, PToV-VIC-ORF1a gene, and SADS-CoV-Cy5-ORF1a gene. The test was applied to 101 clinical swine samples and had an LOD of 100 copies for each virus [37].

The last quadruplex qPCR was designed to identify and differentiate between wild-type (B646L) and gene-deleted ASFV strains (MGF505-2R, EP402R, and I177L). The design was based on the amplification of the p72 and CD2v genes using probes labeled with FAM (B646L), VIC (MGF505-2R), Cy5 (EP402R), and Texas Red (I177L). It was tested on 4239 clinical samples and had a range of LODs between 32.1 and 3.21 copies [38].

PCVs have also attracted interest in the development of multiplex qPCR assays. In this respect, four developments that can simultaneously identify different types of PCVs have been reported. Zou et al. presented a triplex qPCR assay for detecting the PCV2, PCV3, and PCV4 viruses. The test is based on the design of labeled probes to detect the cap genes for PCV2 (Cy5), PCV4 (Texas Red), and rep for PCV3 (FAM). The test was used for analyzing 535 clinical samples and an LOD of 101 copies was obtained [39]. On the other hand, a duplex PCV qPCR was developed by a research group from the Republic of Korea; their proposal was based on amplifying the cap genes of PCV2 and PCV3 using probes labeled with FAM and ROX, respectively. The authors reported an LOD below 50 copies for both viruses [40].

Another research group developed a similar duplex qPCR of PCV from China, and its design comprises the detection of the cap gene for PCV2 and rep for PCV3 using fluorophores FAM and HEX, respectively. The authors reported an LOD of 2.9 copies for PCV2 and 22.5 copies for PCV3, and the authors validated it with 340 samples [41]. Finally, Wang et al. developed a duplex PCV qPCR in which the rep and ORF3 genes were amplified using VIC-labeled probes for PCV2 detection; for PCV3 identification, they designed probes for the rep and cap genes labeled with FAM. This duplex was validated in 336 swine clinical samples and had an LOD of 14 copies for PCV2 and 17 copies for PCV3 [42]. Table 1 summarizes the multiplex qPCR tests, highlighting the targets (virus, gene, and dye) and the LODs.

Several companies have launched multiplex qPCR tests to detect swine viruses (Table 2). For example, IDEXX (Westbrook, Maine, U.S.A.) has three duplex tests that detect PCV2, PCV3, PRRSV1, PRRSV2, PEDV, and PDCoV. Thermo Fisher (Waltham, Massachusetts, U.S.A.) has two tests, one duplex for PRRSV1 and PRRSV2 and a triplex for PEDV, TGEV, and PDCoV. Hermes-Bio (Monterrey, Nuevo León, Mexico) developed three multiplex tests: the first one detects PRRSV, PEDV, and TGEV; the second one detects PoRV-A, PoRV-B, and PoRV-C; and the third one detects PCV2, PRRSV, and PDCoV. Tetracore (Rockville, Maryland, U.S.A.) has two tests: one duplex for PRRSV and influenza A virus (IAV) and the other that consists of a triplex for PEDV, TGEV, and PDCoV.

## 4. Isothermal Methods

The introduction of PCR has allowed for the application of this powerful tool in molecular diagnosis. The age pre- and post-PCR led to a new diagnosis vision through the amplification of genetic material. A thermocycler is necessary for PCR because it regulates temperature changes so that the polymerase can synthesize the genetic material and, as a result, identify the target. This equipment is one of the main limitations of PCR when used in areas without access to sophisticated laboratories or where expensive equipment cannot be purchased.

Molecular alternatives for diagnosis have been sought in light of this scenario and the need to facilitate molecular diagnosis in situations such as pandemics or infection outbreaks in regions of the world where this type of deficiency exists. As a result, isothermal techniques were developed that simplify the detection process by requiring only one incubation at a single temperature. The development of isothermal methods for the diagnosis of porcine viruses is discussed in this section, from loop-mediated isothermal amplification (LAMP) and nucleic acid sequence-based amplification (NASBA) to more recent ones such as recombinase polymerase amplification (RPA) and polymerase spiral reaction (PSR). Figure 2 shows the different isothermal methodologies applied to swine diagnosis and the proportions of these techniques applied to the most extensively researched viruses documented in scientific papers.

Loop-Mediated Isothermal Amplification (LAMP)

There are different isothermal techniques in use today; LAMP, which a Japanese group proposed in 2000, was one of the first to emerge [43]. It is considered one of the most popular isothermal methods and has been used to diagnose a wide variety of pathogens [44,45,46,47]. Two main elements are required for LAMP: (1) the *Bst* polymerase and (2) a set of primers (two forward primers and two reverse primers) that are designed to recognize different regions of the target. These elements are incubated at 60–65 °C for 30 to 60 min to amplify the genetic material; LAMP is compatible with DNA and RNA templates. It is important to note that the use of LAMP has grown significantly in recent times, mainly owing to improvements in the way the results are presented. Traditionally, it was performed using electrophoresis to observe the band pattern, but nowadays colorimetric alternatives use phenol red, which is sensitive to changes in pH when there is a positive sample. There are also alternatives that involve the use of fluorescent dyes that are intercalated in the amplified double strand or that are sensitive to pH changes (after amplification of the target sequence). Fluorescent probes (bound to a quencher) are also used, which emit a fluorescence signal upon releasing the quencher after its hybridization and amplification with its target DNA sequence. These alternative technologies are desirable for use as a point-of-care (POC) system because they enable pathogen identification with the naked eye and in real time [44]. LAMP has the most developed tests regarding the isothermal methods applied for detecting swine viruses, with several dozen published reports. Some of the viruses that LAMP tests can detect are: ASF, PRRSV, PEDV, PCV, IAV, CSFV, getah, PPV, PtoV, SVDV, and PDCoV (Table 3).

Some examples of this technique include the assay developed by Cao et al., who used the LAMP technique in conjunction with carbon dots as biosensors to emit a fluorescence signal caused by a pH change (generated by increasing the amount of pyrophosphoric acid) after amplification of a target region of the P72 gene of ASFV, reporting an LOD of 15.21 copies/µL and a detection time of 32 min for this technique [48]. Similarly, Wang et al. used fluorescence signals to detect ASF; however, they used a probe (binding to the FAM-fluorophore and a quencher) complementary to a region of the 9GL gene of ASFV that harbored a ribonucleotide insertion. Therefore, after hybridization of the probe with the target sequence, ribonuclease H2 cleaves the probe, releasing the quencher and leading to the emission of a fluorescence signal. This assay has a detection limit of 13 copies/µL and provides results in 40 min [49].

Another innovative example is the techniques proposed by Areekit et al., who designed a LAMP duplex in combination with a lateral flow dipstick (LFD). This assay consists of four primer pairs for the simultaneous identification of the conserved regions of the spike gene and an ORF gene of the PEDV and PCV2, respectively. The amplified regions are recognized by specific probes for PEDV and PCV2, generating two purple lines or one (in the absence of PEDV or PCV2) on the strip test line. This technique has a detection limit of 0.1 ng/µL for PEDV and 0.246 ng/µL for PCV2, making it 10 times more sensitive than conventional PCR and R-PCT techniques; the results are obtained in 1.5 h [50].

Nucleic Acid Sequence-Based Amplification (NASBA)

Another isothermal method with more history is the NASBA, which was created in 1991 [51]. It has been widely reported for some time but has not yet been exploited for molecular diagnosis. NASBA has recently become one of the isothermal approaches to emerge in line with isothermal procedures. A set of three enzymes is required to use NASBA: avian myeloblastosis virus (AMV) RT, RNase H, T7 DNA-dependent RNA polymerase, and primers designed according to the target [51]. This mixture is incubated at 41–42 °C for 2 h to amplify the nucleic acids. It is essential to note that NASBA can be used for DNA and RNA. NASBA tests have been recently developed for detecting swine viruses such as CSFV and JEV (Table 3). In this sense, Zhou et al. employed the NASBA technique to detect the NS1 gene of JEV using four pairs of primers and a probe; after its amplification, a fluorescence signal is generated within 10 min. The test has an LOD of six copies per reaction [52].

Moreover, Lu et al. used the NASBA technique to identify CSFV strains (Shimen and HCLV) using probes that, when hybridized with the target sequence, form a G-quadruplex structure that is transformed into a DNAzyme with peroxidase-like activity when bound with hemin, which in the presence of ABTS mediates its oxidation and results in a green color. This technique has a detection limit of 10 copies/mL of CSF viral RNA and the results are generated within 3 h [53].

Recombinase Polymerase Amplification (RPA)

An isothermal method of great interest is RPA, which was developed in 2006 [54]. As a result of its ease of use, RPA is one of the most developed procedures. To enable the amplification of the target, this technique requires a set of enzymes: recombinase, DNA polymerase, and a single-stranded DNA binding (SSB) protein [54]. The temperature range in which the target amplification reaction occurs is 37–42 °C and the reaction takes 30 to 60 min. Like other techniques, RPA is compatible with DNA and RNA. In addition, the design of primers is very similar to those designed in a traditional PCR, which is advantageous considering the number of primers (four) needed for LAMP.

Research groups have recently developed several alternatives to electrophoresis, which—as well as LAMP—is the traditional approach to reveal the result of RPA amplification. Among some alternatives is the use of fluorescent dyes for real-time detection, which involves the use of probes (attached to a fluorophore and a quencher) that hybridize with the target sequence. Following the amplification by the polymerase of the target DNA, the probe is cleaved, releasing the fluorophore and quencher and emitting a fluorescence signal [55]. Other alternatives are the lateral flow assay (LFA) or immunostrips, which detect the presence of amplicons labeled with biotin and the fluorescent probe previously generated by the RPA reaction in the presence of the target virus and that are recognized by anti-biotin or anti-fluorophore antibodies on the immunostrip test line [56]. These types of alternatives provide real-time visualization of the results [55].

RPA has been widely used to identify porcine viruses such as ASF, PRRSV, PCV, PDCoV, PEDV, SADS-CoV, and porcine encephalomyocarditis virus (EMCV) (Table 3). Regarding this technique, Wang et al. performed two assays: a real-time reverse transcription-RPA (RT-RPA) and a lateral flow biosensor (LFB) in combination with RT-RPA. These assays were able to identify 13 different EMCV strains using a pair of primers and a probe to amplify and hybridize, respectively, conserved regions of the 3D gene. In the presence of the virus, a fluorescence signal is emitted by the qRT-RPA assay, whereas the LBF RT-RPA shows a red line on the test line. Both assays generate results within 20 min, with LODs of 1.0 × 10^2^ and 1.0 × 10^1^ copies for the qRT-RPA and LFB RT-RPA, respectively [57].

On the other hand, Cong et al. used the RT-RPA technique to detect SADS-CoV using a set of pairs of primers and probes (labeled with FAM) capable of amplifying conserved regions of the M gene of the virus, generating a signal of fluorescence in 15 min with an LOD of 74 copies/µL [58].

Polymerase Spiral Reaction (PSR)

One of the most recent isothermal techniques is the PSR, which was first reported in 2015 [59]. This method is based on using the *Bst* enzyme and a couple of primers designed for the target. For target amplification, it is necessary to incubate at 61–65 °C for 45 to 60 min. The method’s name comes from the fact that the amplification reaction will produce a spiral structure if the target is present [59]. For detection purposes, fluorescent dyes or a turbidimeter can be used to measure the fluorescence or detect turbidity, respectively. Although it is a relatively new technique, there are some PSR proposals for detecting porcine viruses including ASF, PCV, and PEDV. Table 3 shows the main characteristics of the reported tests based on isothermal methods that are used for porcine virus detection. Wozniakowski et al. reported a novel variant of the PSR called the polymerase cross-linking spiral reaction (PCLSR) for detecting ASV in the blood of pigs and wild boars via the use three pairs of primers. The first pair of primers are external and complementary to the 3′ sequence of the p72 gene, the second pair of primers are complementary to the exogenous sequence of the black widow alpha-latrotoxin gene, and the last pair are ASFV-specific cross-linking primers. This test generates results in 45 min through a fluorescence signal and exhibits an LOD of 7.2 × 10^2^ copies/µL [60].

Similarly, Wang et al. developed an alternative reverse transcription PSR (RT-PSR) that was capable of detecting the ORF3 gene of PEDV in porcine tissue samples using primer pairs that form a spiral after amplification of this sequence; this causes a color change to the naked eye using phenol red and cresol red dyes in a time of 50 min [61].

**Table 3 vetsci-10-00609-t003:** Isothermal tests developed for the detection of swine viruses describing the involved enzymes, the incubation conditions for the reactions, the viral targets, the LODs, the detection times, and the detection methods.

Isothermal Method	Enzymes	Conditions	Target	Detection Signal	LOD	Detection Time (min)	Reference
LAMP	*Bst*	25–60 min at 60–65 °C	ASFV	Fluorescence (Carbon nanodots)	15.21 copies/µL	30	[48]
			ASFV	Fluorescence	330 copies/µL	25	[62]
			ASFV	Fluorescence (FAM)	13 copies/µL	40	[49]
			PRRSV	Fluorescence (Picogreen)	80 fg/µL	40	[63]
			PRRSV	Fluorescence and precipitate formation (Ethidium bromide, Picogreen and magnesium pyrophosphate precipitate)	1 × 10^0^ to 1 × 10^1^ copies/reaction	70	[64]
			PRRSV	Colorimetric (HBN)	10^3^ copies/reaction	60	[65]
			PRRSV	Turbidity	0.1 TCID_50_	62	[66]
			PRRSV	Fluorescence (SYBR)	0.01 ng/µL	60	[67]
			PRRSV	Fluorescence (SYBR)	0.001 TCID_50_	50	[68]
			PRRSV	Electrophoresis	5 copies/tube	47	[69]
			PRRSV	Turbidity and fluorescence (SYBR)	10^2^ to 10^4^ TCID_50_/mL	50 to 60	[70]
			PRRSV	Colorimetric (HBN)	0.1 to 1 TCID_50_/0.1 mL	40	[71]
			PCV1	Turbidity	10 copies/µL	62	[72]
			PCV3	Fluorescence SYTO-9	1 × 10^1^ copies/µL	70	[73]
			PCV2- PEDV	Lateral flow dipstick (LFD)	0.246 ng/µL for PCV2 and 0.1 ng/µL for PEDV	90	[50]
			PCV3	Fluorescence (FAM)	50 copies/reaction	17.34 ± 4.45	[74]
			PEDV	Fluorescence	2 × 10^0^ TCID_50_/_mL_ to 2.8 × 10^1^ TCID_50_/mL	50	[75]
			PEDV	Fluorescence (SYBR)	0.0001 ng/µL	62	[76]
			IAV	Lateral flow dipstick (LFD)	7.8 pg/µL	30	[77]
			CSFV	Colorimetric (HBN)	100 copy numbers	60	[78]
			PDCoV	Fluorescence (FAM)	25 copies/µL	<40	[79]
			PDCoV	Fluorescence (SYBR)	1 × 10^1^ copy numbers	70	[80]
			PPV	Electrophoresis	12 fg	45	[81]
			Getah	Fluorescence (SYBR)	2.61 copies/µL	50	[82]
			PToV	Fluorescence (SYBR)	1 × 10^1^ copies/μL	70	[83]
			SVDV	Fluorescence (SYBR)	50 copies per assay	30 to 60	[84]
NASBA	AMV-RT, RNase H, and T7 polymerase	120 min at 41–42 °C	CSFV	Fluorescence (ThT)	2 copies/µL	120	[85]
			JEV	Fluorescence (FAM)	6 copies/reaction	10 to 50	[52]
			CSFV	Colorimetric (ABTS)	10 copies/mL	180	[53]
RPA	Recombinase, DNA polymerase, and SSB	10–60 min at 37–42 °C	ASFV	Fluorescence (FAM)	93.4 copies/reaction	16	[86]
			ASFV	Lateral flow dipstick (LFD)	150 copies/reaction	10	[56]
			EMCV	Fluorescence (FAM) and lateral flow dipstick (LFD)	1 × 10^2^ copies for fluorescent RPA and 1 × 10^1^ copies for LFD	20	[57]
			PCV2	Fluorescence (FAM) and lateral flow dipstick (LFD)	10^2^ copies/reaction	20	[87]
			PCV2	Electrophoresis	10^2^ copies	~30	[88]
			PDCoV	Fluorescence (FAM)	100 copies/reaction	20	[89]
			PDCoV	Lateral flow dipstick (LFD)	1 × 10^2^ copies/µL	10	[90]
			PEDV	Lateral flow dipstick (LFD)	1 × 10^2^ copies/µL	30	[91]
			PEDV	Lateral flow dipstick (LFD)	10^2^ TCID_50_/mL	25	[92]
			PRRSV	Lateral flow dipstick (LFD)	5.6 × 10^−1^ TCID_50_	30	[93]
			SADS-CoV	Fluorescence (FAM)	74 copies/µL	30	[58]
PSR	*Bst*	61–65 min at 45–60 °C	ASFV	Fluorescence (SYBR)	7.2 × 10^2^ copies/µL	45	[60]
			PEDV	Colorimetric (Phenol red and cresol red)	1 fg/mL	50	[61]
			PCV3	Colorimetric (Phenol red and cresol red)	1.13 × 10^2^ copies/µL	50	[94]

Note: Loop-mediated isothermal amplification (LAMP), Nucleic acid sequence-based amplification (NASBA), Recombinase polymerase amplification (RPA), Polymerase spiral reaction (PSR), Avian myeloblastosis virus (AMV), Single-stranded DNA binding (SSB) protein, African swine fever virus (ASFV), Porcine reproductive and respiratory syndrome virus (PRRSV), Porcine circovirus (PCV), Porcine epidemic diarrhea virus (PEDV), Influenza A virus (IAV), Classical swine fever virus (CSFV), Porcine delta coronavirus (PDCoV), Porcine parvovirus (PPV), Porcine torovirus (PtoV), Swine vesicular disease virus (SVDV), Japanese encephalitis virus (JEV), Swine acute diarrhea syndrome coronavirus (SADS-CoV), TCID50: 50% tissue culture infective doses.

## 5. Novel Methods: CRISPR-Cas and Microfluidics Platforms

CRISPR is considered one of the newest and most versatile methods; numerous CRISPR variants have been developed, but the technique’s fundamental principles still involve using complex RNA sequences, which are designed to detect the target, and Cas nuclease, which consists of a wide range of nuclease variants such as Cas9, Cas13a, Cas12a, Cas12b, and Cas14 [95]. In addition, for molecular diagnosis, a single-stranded DNA sequence labeled with a fluorophore and quencher is usually used as a reporter [96].

Based on CRISPR-Cas, research groups are constantly creating new techniques and combining existing ones. For instance, researchers have combined this technique with LAMP and RPA. For example, Qin et al. developed CRISPR-RPA and CRISPR-LAMP systems to identify ASFV that consist of the use of a CRISPR RNA (crRNA), Cas 12a, and a single-stranded DNA reporter (ssDNA) that has a fluorophore and a quencher. After amplifying the ASFV region of interest via RPA or LAMP, the crRNA sequence guides Cas 12a to the target region, releasing a fluorescence signal generated by the reporter ssDNA (Figure 3). CRISPR-RPA has an LOD of 7 × 10^3^ copies/µL and a detection time of 40 min, whereas CRISPR-LAMP has an LOD of 5.8 × 10^2^ copies/µL and generates results in 60 min [96]. CRISPR-Cas multiplex tests have also been developed with the capability of being analyzed by colorimetry, fluorescence, or immunostrips (LFD) [95]. CRISPR-Cas-based tests for the detection of swine viruses have mainly been reported for the identification of ASFV, with a small number of reports regarding the detection of PRRSV, PCV, PEDV, TGEV, PDCoV, and SADS-CoV. For example, Liu et al. designed a CRISPR/Cas 12a platform in conjunction with RT-LAMP that was capable of simultaneously identifying PEDV, TGEV, PDCoV, and SADS-CoV via colorimetric detection and obtaining results in less than an hour; this platform exhibited single-copy sensitivity [97]. Table 4 shows the key features of the reported tests based on CRIPSR-Cas technology in combination with some methodologies, such as RPA or LAMP, to identify porcine viruses.

Microfluidics has developed tremendously and has great potential in molecular diagnostics, mainly due to the considerable reduction of reagents, cost savings, and faster processing times for test results. The potential of microfluidics is also ascribed to the ability to have complete laboratories on a chip, which enables the performance of complex analyses in the palm of a hand and the implementation of numerous parallel experiments in a condensed area [107].

The demand for low-cost, fast, simple-to-use, remote, and sensitive molecular diagnostic tests that can be used anywhere has become more pronounced, especially in the wake of the recent pandemic. In this scenario, microfluidics might offer the characteristics that massive molecular diagnosis requires [108].

Currently, a wide variety of microfluidics tests have been developed to address human health [109], animal health [110], and food safety [111] issues, among others. There are some interesting proposals in the specific case of porcine virus detection. One of them is a multiplex PCR microfluidics system that allows the detection of PRRSV, PEDV, PRV, and PCV2 in clinical samples on site. This chip consists of eight chambers, each containing a PCR reaction, making it possible to identify the presence of viruses from eight samples simultaneously in one hour and with an LOD of 1 copy/µL [112]. Other innovative qPCR-based microfluidics systems are portable devices that use magnetofluid drops to purify DNA from saliva, blood, or tissue. The PCR reaction is performed within a cartridge to identify ASFV, generating results through a fluorescence signal in less than 30 min and with LODs ranging from 10^3^ copies/µL to 2 copies/µL [113].

Three-dimensional printed microfluidic chips, based on RT-LAMP, have also been proposed for multiple detection of PEDV, TGEV, and PDCoV within 30 min by emitting a fluorescence signal, with LODs of 10 copies per reaction for PEDV and PDCoV and 100 copies per reaction for TGEV [114]. Table 5 shows the microfluidics platforms that have been applied for detecting porcine viruses such as ASF, PRRSV, PEDV, and PCV, among others.

## 6. Current Challenges and Future Recommendations

The swine industry has ongoing challenges for sustaining the highest productive parameters while fighting against the great diversity of pathogens (viruses, bacteria, protozoa, fungi, and parasites) [118]. There are several significant reasons why strict biosecurity protocols, pathogen monitoring, and surveillance are necessary, such as the high mutation rate of viruses such as PRRSV or IAV, the exponential capacity of viral replication in pigs, subclinical infections, the increased mobility of personnel on farms, the arrival of vehicles and supplies, and the natural reservoirs of pathogens [119,120].

Diagnostic tests are the primary tools for detecting and preventing diseases, determining health status, tracking seroprevalence, epidemiological surveillance, and understanding pathogen dynamics inside and outside farms [121]. Detecting viral flow and antibodies within the farm or identifying the strains affecting the herd could aid in understanding and evaluating the effectiveness of management interventions to reduce disease impacts [122]. One of the most profitable investments in the swine industry is in diagnostic testing and monitoring programs, because the economic repercussions of failing to diagnose can be catastrophic for a farm [123]. The idea is simple: only what is measured can be improved, and diagnosis is the instrument for measuring diseases on farms.

The swine industry has tests for detecting antibodies by ELISA, pathogens by qPCR (simple or multiplex), and pathogen strains by sequencing [121]. However, to implement these diagnostic methodologies or emerging ones, they must be tailored to the needs and characteristics of the porcine industry. Despite the existence of portable, sensitive, and state-of-the-art diagnostic technologies (such as CRISPR-Cas, isothermal, etc.), their implementation and use in the field must meet technical, economic, operational, and logistical considerations [124]. Finally, we will discuss the main challenges of the new diagnostic technologies for implementation in the swine industry; in addition, Table 6 compares the main characteristics of the different types of molecular tests.

Price. The price of tests for analyzing animal health is between 50 and 70% cheaper than for human health. The average price of qPCR tests for the swine industry is around USD 22–35. If the technology is prohibitively expensive, it might not be appropriate for the animal industry, even if it meets all the technical requirements. Price is critical to maintaining ongoing diagnosis and monitoring.Turnaround time. Turnaround times for qPCR tests are 1–6 days. The processing time is a key parameter, since the capacity for action on the farm increases with the rapidity and accessibility of the test.Reliability. Tests must be sensitive and specific; for example, qPCR tests can detect 100 to 1 copies/mL. This range can be used as a reference for new technologies that seek to be implemented [125].Supply Chain. Suppliers of these technologies must be prepared to offer mass production, guarantee low prices, and maintain a continuous supply. The supply chains for the newest technologies are typically not prepared; as a result, the supply is complicated and expensive [126].Import costs. Local or regional suppliers should be identified to avoid excessive import and distribution costs [127].No cold chain. New emerging technologies should avoid refrigerated logistics since they increase the product’s costs and reduce the viability and lifetime [126].Easy to perform and interpret. The ideal technologies would be on-site systems that are fast and easy to process without requiring specialized technicians to perform the procedure and interpret the results [128].Available infrastructures. The necessary equipment and facilities must be available to perform the methodologies or implement the new technologies on farms or in local laboratories.Controls to avoid false results. New technologies require controls to avoid false positives and controls to detect when the result is a false negative.Detection of multiple pathogens. The diagnostic process includes sample collection, shipment, analysis, and the results report; thus, it is a significant effort and investment to detect only one pathogen. New technologies that seek to be implemented in the sector should be able to detect at least three pathogens in the same test to support issues of differential diagnosis, monitoring, and surveillance [129].

## 7. Conclusions

The pork industry plays a crucial role in the nutrition of the world population; it is also an essential component for human health since pigs are an ideal study model due to their remarkable genetic and physiological similarities to humans. The similarity to humans justifies the associated zoonotic risks, as pigs are potential reservoirs of pathogens such as hepatitis E virus and swine influenza virus. Thus, disease monitoring programs for pigs are vital to avoid pandemics or infectious outbreaks in humans. This review covers the main viruses affecting the swine industry (ASFV, PRRSV, PEDV, and PCV), their transmission routes, their symptoms, and the main diagnostic tests used. Furthermore, molecular tests stand out, such as the gold standard qPCR, which, despite being multiplex and providing information on concurrent infections affecting pigs, has significant limitations that render it unsuitable. For example, these assays require qualified personnel, prior sample processing, refrigerated reagents, and well-equipped laboratories that are typically located in cities rather than in rural areas. Additionally, isothermal tests (LAMP, RPA, PSR, and NASBA) are highlighted; these tests can be combined with other techniques such as CRISPR-Cas or microfluidics platforms to provide faster results (less than an hour) with low LOD values. Their remarkable sensitivity allows them to detect a single viral copy without the need for expensive and sophisticated equipment or highly qualified personnel. In addition, it is important to note that these molecular biology techniques require specific temperatures and incubation times to detect the target virus, except for the lateral flow dipstick (LFD), since this can be performed at room temperature without the need to use a water bath. Finally, the challenges to be overcome were discussed, which is crucial for the future development of tests that can be used in any location and at any temperature (without the need for a water bath). Future developments must allow inexpensive, rapid, and early diagnosis (times less than hour) tests for daily use that help make instant decisions to avoid significant economic losses or the development of future pandemics that affect humanity.

## Figures and Tables

**Figure 1 vetsci-10-00609-f001:**
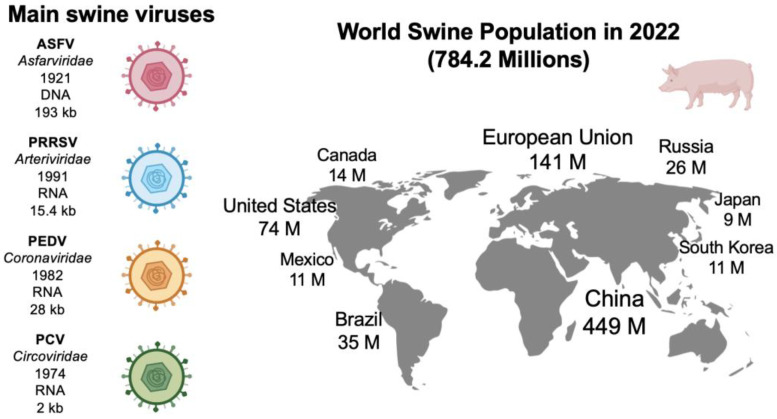
Viruses with the most significant impact in the pig industry, highlighting the main characteristics. Pig population and its geographical distribution, referenced from the Our World in Data database [2]. Note: African swine fever virus (ASFV), porcine reproductive and respiratory syndrome virus (PRRSV), porcine epidemic diarrhea virus (PEDV), porcine circovirus (PCV).

**Figure 2 vetsci-10-00609-f002:**
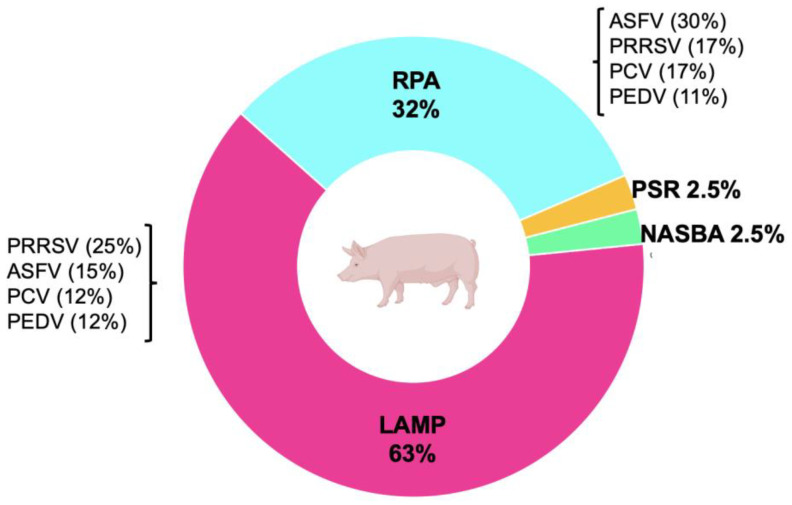
The proportion of isothermal tests developed to identify porcine viruses that have been published in scientific papers using the NCBI database. In addition, displayed are the most extensively studied viruses in the cases of LAMP and RPA. Note: African swine fever virus (ASFV), Porcine reproductive and respiratory syndrome virus (PRRSV), Porcine epidemic diarrhea virus (PEDV), Porcine circovirus (PCV), Loop-mediated isothermal amplification (LAMP), Nucleic acid sequence-based amplification (NASBA), Recombinase polymerase amplification (RPA), Polymerase spiral reaction (PSR).

**Figure 3 vetsci-10-00609-f003:**
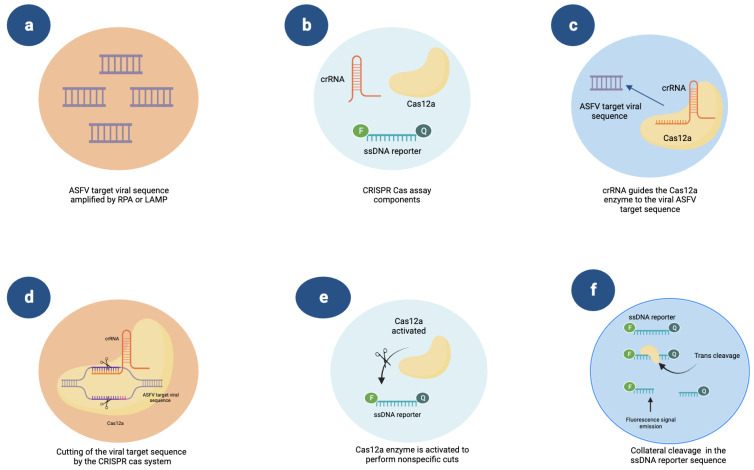
Schematic illustration of the CRISPR-Cas system’s mechanism of action using ssDNA (single-stranded DNA) as a reporter for the detection of ASFV. (**a**) The ASFV DNA target sequence is amplified by the RPA or LAMP isothermal assays. (**b**) Components required for ASFV detection by the CRISPR-Cas system, which are CRISPR-RNA = crRNA, the enzyme Cas12a, and the ssDNA reporter labeled with a fluorophore (F) and a quencher (Q). (**c**) The Cas12a enzyme is guided by the crRNA to the target viral sequence. (**d**) The ASFV DNA sequence is cut by the CRIPR-Cas system. (**e**) After cutting the target sequence, the Cas12a enzyme is activated to perform non-specific ssDNase activity. (**f**) The non-specific cut in the ssDNA reporter sequence results in the release of the quencher, with a fluorescence signal indicating the presence of ASFV.

**Table 1 vetsci-10-00609-t001:** Multiplex qPCR tests reported in the literature, with their respective targets (virus, gene, and dye) and limits of detection (LODs).

Type of Multiplex Test	Targets and Reporters (Virus-Gene-Dye)	LOD (Viral Copies)	Reference
Quadruplex RT-qPCR	(1) PEDV-N-JUN (2) PDCoV-N-ABY (3) TGEV-N-VIC (4) SADS-CoV-N-FAM	4–16	[22]
Quadruplex RT-qPCR	(1) PEDV-N-JOE (2) PDCoV-M-FAM (3) TGEV-N-Texas Red (4) SADS-CoV-N-Cy5	121	[36]
Quadruplex RT-qPCR	(1) PEDV-ORF1a-FAM (2) PDCoV-ORF1b-Texas Red (3) PtoV-ORF1a-VIC (4) SADS-CoV-ORF1a-Cy5	100	[37]
Quadruplex qPCR	ASFV strains (1) B646L-p72-FAM (2) MGF505-2R-CD2v-VIC (3) EP402R-CD2v-Cy5 (4) I177L-CD2v-Texas Red	3.21–32.1	[38]
Triplex RT-qPCR	(1) ASFV-p72-FAM (2) CSFV-5′ UTR-VIC (3) PRRSV-ORF7-Cy5	1.78	[34]
Triplex RT-qPCR	(1) ASFV-p72-Texas Red (2) CSFV-5′ UTR-JOE (3) APPV-5′ UTR-FAM	2.52	[35]
Triplex qPCR	(1) PCV2-cap-Cy5 (2) PCV3-rep-FAM (3) PCV4-cap-Texas Red	101	[39]
Duplex qPCR	(1) PCV2-cap-FAM (2) PCV3-cap-ROX	50	[40]
Duplex qPCR	(1) PCV2-cap-FAM (2) PCV3-rep-HEX	2.9–22.5	[41]
Duplex qPCR	(1) PCV2-rep/cap-VIC (2) PCV3-rep/ORF3-FAM	14–17	[42]

Note: Real-time PCR (qPCR), Reverse transcription polymerase chain reaction (RT-PCR), Porcine epidemic diarrhea virus (PEDV), Porcine delta coronavirus (PDCoV), Transmissible gastroenteritis virus (TGEV), Swine acute diarrhea syndrome coronavirus (SADS-CoV), Porcine torovirus (PtoV), African swine fever virus (ASFV), Classical swine fever virus (CSFV), Porcine reproductive and respiratory syndrome virus (PRRSV), Atypical porcine pestivirus (APPV), Porcine circovirus (PCV).

**Table 2 vetsci-10-00609-t002:** Multiplex qPCR tests developed by companies, with their respective targets (virus and dye).

Type of Multiplex Test	Targets and Reporter (Virus-Dye)	Detection Time (min)	Company (Web)
Duplex qPCR	(1) PCV2-FAM (2) PCV3-Cy5	90	Idexx.com
Duplex RT-qPCR	(1) PRRSV1-Cy5 (2) PRRSV2-FAM	90	Idexx.com
Duplex RT-qPCR	(1) PEDV-FAM (2) PDCoV-Cy5	90	Idexx.com
Duplex RT-qPCR	(1) PRRSV1-VIC (2) PRRSV2-FAM	90	Thermofisher.com
Triplex RT-qPCR	(1) PEDV-LIZ (2) TGEV-FAM (3) PDCoV-VIC	90	Thermofisher.com
Triplex RT-qPCR	(1) PRRSV-FAM (2) TGEV-VIC (3) PEDV-Cy5	90	Hermes-Bio.com
Triplex RT-qPCR	(1) PCV2-FAM (2) PRRSV-VIC (3) PDCoV-Cy5	90	Hermes-Bio.com
Triplex RT-qPCR	(1) PoRV-A-FAM (2) PoRV-B-VIC (3) PoRV-C-Cy5	90	Hermes-Bio.com
Duplex RT-qPCR	(1) PRRSV-N.A. (2) IAV-N.A.	140	Tetracore.com
Triplex RT-qPCR	(1) PEDV-FAM (2) TGEV-N.A. (3) PDCoV-Cy5	140	Tetracore.com

Note: Real-time PCR (qPCR), Reverse transcription polymerase chain reaction (RT-PCR), Porcine circovirus (PCV), Porcine reproductive and respiratory syndrome virus (PRRSV), Porcine epidemic diarrhea virus (PEDV), Porcine delta coronavirus (PDCoV), Transmissible gastroenteritis virus (TGEV), Porcine rotavirus (PoRV), Influenza A virus (IAV).

**Table 4 vetsci-10-00609-t004:** Recent advancements in tests based on CRIPSR-Cas technology describe the enzymes involved, the reaction incubation conditions, the viral targets, and the detection methods.

Method Combined with CRISPR-Cas	Enzymes	Conditions	Target	Detection Signal	LOD	Detection Time (min)	Reference
LAMP	*Bst*, Cas12a	▪ 40 min at 65 °C (LAMP Reaction) ▪ 10 min at 37 °C (CRISPR reaction)	PEDV, TGEV, PDCoV, and SADS-CoV	Fluorescence (ROX)	1 copy/µL	50	[97]
LAMP	*Bst*, Cas12a	▪ 40 min at 65 °C (LAMP reaction) ▪ 20 min at 37 °C (CRISPR reaction)	ASFV	Fluorescence (FAM)	5.8 × 10^2^ copies/µL	60	[96]
LAMP	*Bst*, Cas12a	▪ 30 min at 65 °C (LAMP reaction) ▪ 10 min at 37 °C (CRISPR reaction)	ASFV	Fluorescence (SYTO 9)	1 copy/µL	40	[98]
LAMP	*Bst*, Cas12a	▪ 30 min at 63 °C	PCV2	Fluorescence (FAM)	1 copy/µL	30	[99]
RPA	Cas12a, recombinase, DNA polymerase, and SSB	▪ 40 min at 37 °C	ASFV	Fluorescence (FAM)	7.4 × 10^4^ copies/µL	40	[96]
RPA	Cas12a, recombinase, DNA polymerase, and SSB	▪ 20 min at 39 °C (RPA reaction) ▪ 15 min at 37 °C (CRISPR reaction)	ASFV	Fluorescence (HEX)	1.16 copies/µL	35	[100]
RPA	Cas12a, recombinase, DNA polymerase, and SSB	▪ 20 min at 39 °C (RPA reaction) ▪ 15 min at 37 °C (CRISPR reaction)	ASFV	Fluorescence (FAM)	2 copies of DNA/reaction	35 min	[101]
RPA	Cas12a, recombinase, DNA polymerase, and SSB	▪ 90 min at 39 °C (RPA reaction) ▪ 30 min at 37 °C (CRISPR reaction)	ASFV	Lateral flow dipstick (LFD)	2 × 10^2^ copies of viral genome	120	[102]
RPA	Cas13a, recombinase, DNA polymerase, and SSB	▪ 80 min at 37 °C	PCV4	Lateral flow dipstick (LFD)	1 × 10^0^ to 1 × 10^1^ copies/µL	80	[103]
RPA	Cas12a, recombinase, DNA polymerase, and SSB	▪ 25 min at 37 °C	PRRSV	Fluorescence (FAM)	1 × 10^0^ copies/µL	25	[104]
RPA	Cas13a, recombinase, DNA polymerase, and SSB	▪ 60 min at 37 °C	PRRSV	Fluorescence (FAM)	1.72 × 10^2^ copies/µL	60	[105]
LFD	Cas12a	▪ 30 min at 37 °C	ASFV	Lateral flow dipstick (LFD)	2 × 10^1^ copies/reaction	30	[106]

Note: Loop-mediated isothermal amplification (LAMP), Recombinase polymerase amplification (RPA), Lateral flow dipstick (LFD), Single-stranded DNA binding (SSB) protein, Porcine epidemic diarrhea virus (PEDV), Transmissible gastroenteritis virus (TGEV), Porcine delta coronavirus (PDCoV), Swine acute diarrhea syndrome coronavirus (SADS-CoV), African swine fever virus (ASFV), Porcine circovirus (PCV), Porcine reproductive and respiratory syndrome virus (PRRSV).

**Table 5 vetsci-10-00609-t005:** Microfluidics-based tests reported in the literature, detailing the enzymes involved, the reaction incubation conditions, the viral targets, and the detection methods.

Microfluidics Platform	Enzymes	Conditions	Target	Detection	Reference
Microfluidics Multiplex-PCR	DNA polymerase	▪ 15 min at 50 °C (Reverse transcription) ▪ 3 min at 95 °C (Denaturation) ▪ 40 cycles: 15 s at 95 °C/45 s at 55 °C (Denaturation and amplification)	PRRSV, PEDV, PRV, and PCV2	Fluorescence (SYBR)	[112]
Magnetofluidics Device-qPCR	DNA polymerase	▪ 3 min at 95 °C (Denaturation) ▪ 50 cycles: 5 s at 95 °C/30 s at 58 °C (Denaturation and amplification)	ASFV	Fluorescence	[113]
3D-printed Microfluidics Device-LAMP	*Bst*	30 min at 65 °C	PEDV, TGEV, and PDCoV	Fluorescence EvaGreen	[114]
Handheld Microfluidics Chip-LAMP	*Bst*	60 min at 65 °C	PEDV, TGEV, PoRV, and PCV2	Colorimetric phenol red	[115]
Microfluidics Multiplex-LAMP	*Bst*	60 min at 63.5 °C	ASFV, PPV, PCV2, PRV, and PRRSV	Fluorescence (SYBR)	[116]
Microfluidics Multiplex-LAMP	*Bst*	40 min at 63.5 °C	PEDV, PDCoV, and SADS-CoV	Fluorescence (SYBR)	[117]

Note: Real-time PCR (qPCR), Loop-mediated isothermal amplification (LAMP), Recombinase polymerase amplification (RPA), Porcine reproductive and respiratory syndrome virus (PRRSV), Porcine epidemic diarrhea virus (PEDV), Porcine pseudorabies virus (PRV), Porcine circovirus (PCV), African swine fever virus (ASFV), Transmissible gastroenteritis virus (TGEV), Porcine delta coronavirus (PDCoV), Porcine rotavirus (PoRV), Porcine parvovirus (PPV), Swine acute diarrhea syndrome coronavirus (SADS-CoV).

**Table 6 vetsci-10-00609-t006:** Comparative analysis of different molecular technologies for the diagnosis of swine viruses.

Type of Test	Price	Specificity	Sensibility	Response Time	Available Infrastructure	Detection of Multiple Pathogens	Operation Difficulty
Point of Care (POC)	💲	🎯	🔍	⌛	⚙	🦠	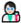
qPCR	💲💲	🎯🎯🎯🎯🎯	🔍🔍🔍🔍🔍	⌛⌛	⚙⚙⚙⚙	🦠	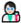 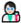
qPCR Multiplex	💲💲💲	🎯🎯🎯🎯🎯	🔍🔍🔍🔍🔍	⌛⌛	⚙⚙⚙⚙	🦠🦠🦠	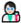 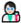
Sequencing	💲💲💲💲💲	🎯🎯🎯🎯🎯	🔍🔍🔍🔍🔍	⌛⌛⌛⌛⌛	⚙	🦠🦠🦠🦠🦠	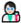 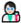 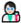 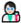 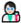

Note: The images indicate a scale from 1 to 5, with 1 representing the minimum and 5 indicating the maximum.

## Data Availability

No new data were created or analyzed in this study. Data sharing is not applicable to this article.

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
