# Peer review of "Emergent Molecular Techniques Applied to the Detection of Porcine Viruses"

_vetsci, 2023, doi:10.3390/vetsci10100609_

Round 1

Reviewer 1 Report

The review summerized the main viruses in pig industry (ASFV,PEDV, PRRSV, PCV) and the detection methods that have been used in pig industry and laboratory. The review is nice for the field of pig viruses. But there are some prospects need to be improve: 1. The lauguage need to be modified intensively; 2. the table which list the detection methods such as Table 2, Table 3 and Table4, the detailed information including LOD and detection time need to show clearly, because the point of care detection, especially in pig industry, the time spend and the lowest copies of virus can be detected is important for the method selection and the virus control. early detetion of virus is the key step. 

Most of the lauguage is acceptable. But authors need to examine the manuscirpt sentence by sentence. For example the line 22-24, must overcome to develop....

Author Response

Dear Reviewer,

Thank you!!

Reviewer 2 Report

In this manuscript titled “Emergent molecular techniques applied to the detection of porcine viruses”, the application of various molecular diagnostic techniques in the pig industry, including qPCR and isothermal methods, was investigated. Additionally, the manuscript extensively discussed emerging technologies such as CRISPR-Cas and microfluidic platforms. The primary viruses impacting pig health were identified, and the challenges in developing cost-effective, accessible, rapid, and user-friendly molecular diagnostic tests were described in the market. The manuscript is well-organized and has certain significance. However, there are still a lot of problems in this manuscript that need to be revised, I would suggest accepting it after the following concerns are addressed.

1.         The language needs considerable attention.

2.         In line 47, it is mentioned that "mainly in swine farms." This means that swine farms are the primary locations where the outbreak is predominantly observed. However, it does not imply that swine farms are the only major locations or that this is true in all regions globally. Whether there are research studies to support this claim may require further investigation and analysis.

3.         Please pay attention to writing format issues, such as proper capitalization and spacing. For example, in line 48, promote should be changed to promote; In line 97, for is changed to in; For line 121, replace vacuum with vacuum; In line 134 is changed to on; Add the before porcine in line 148; Line 206 and removed; Line 306 is removed; On line 527, change qPCR tests to qpcr tests.

4.         The layout of having a single sentence as a paragraph in line 73 is not very logical.

5.         The logic and paragraph structure in lines 75 and 80 are inconsistent.

6.         The harmful effects and impacts of viruses on society can be omitted in the section "Main swine viral diseases" as they have already been discussed in the introduction and illustrated in Figure 2. It is necessary to identify the main emphasis of the article and improve its logical consistency.

7.         The content from line 128 to 131 lacks references.

8.         In line 178, "In recent years," and in line 180, "almost four decades ago," for the sake of accuracy, it would be helpful to provide more specific time ranges.

9.         In order to enhance credibility and relevance, it would be beneficial to provide specific examples or instances to support the description of importance and certain features in the manuscript. Additionally, including more details about ongoing research or practical applications can further increase the credibility and relevance of the information.

10.      In sentence 324, the author mentions the amplification of targets using fluorescent dyes, but then refers to these techniques as favorable for "Point of care (POC) system." The relationship and connection between these two concepts are not clear. The author can provide a better explanation of the application and advantages of LAMP technology in POC systems.

11.      In line 375, it is mentioned that alternative methods to gel electrophoresis have been developed, but it is not specified how these methods are applied for the visualization of RPA results.

12.      The first row of Table 4 has a formatting error.

13.      The small black square boxes in the third row of the second and third columns of Table 5 should be explained in the caption of the figure to indicate their meaning.

Author Response

Dear Reviewer,

Thank you!!

Reviewer 3 Report

This review summarizes all the available diagnostic methods for porcine viruses. It is well written and informative. However, some major concerns have to be further addressed.

Major comment:

1.       References should be included in some sentences: Line 110-111, line 114-118, line 128-132, line 133-134, line 148-150, line 155, line 164-166, line 201-204.

2.       Ideas in some sentences are complicated so it is highly recommended to rephrase sentences: line 97-99, line172-173, line 568-583.

3.       It is highly recommended to unify all the LOD value. Some presents copies/ul while some is only copy number.

4.       Figure 3 does not clearly illustrate the working principle of CRISPR for the diagnostic purpose. The figure only explains how the crRNA orientates Cas proteins to the newly amplified DNA, but the usage of ssDNA reporter is not clearly explained or linked to this process. It is recommended to take a porcine virus as an example to explain this method.

It is highly recommended to do the proof-reading carefully. There are some grammatical errors.

Author Response

Dear Reviewer,

Thank you!!

Round 2

Reviewer 2 Report

no other comments

Author Response

Thank you very much for your valuable feedback and observations.